# Having a Break or Being Imprisoned: Influence of Subjective Interpretations of Quarantine and Isolation on Boredom

**DOI:** 10.3390/ijerph19042207

**Published:** 2022-02-15

**Authors:** Silke Ohlmeier, Corinna Klingler, Isabell Schellartz, Holger Pfaff

**Affiliations:** 1Institute of Medical Sociology, Health Services Research and Rehabilitation Science, University of Cologne, 50933 Cologne, Germany; holger.pfaff@uk-koeln.de; 2Faculty of Health Sciences Brandenburg, University of Potsdam, 14476 Potsdam, Germany; corinna.klingler@uni-potsdam.de; 3Institute of Health Care Research, Rhineland State Council, LVR-Institut für Versorgungsforschung, 51109 Cologne, Germany; ischella@web.de

**Keywords:** boredom, COVID-19, quarantine, social constructivism

## Abstract

Boredom has been identified as one of the greatest psychological challenges when staying at home during quarantine and isolation. However, this does not mean that the situation necessarily causes boredom. On the basis of 13 explorative interviews with bored and non-bored persons who have been under quarantine or in isolation, we explain why boredom is related to a subjective interpretation process rather than being a direct consequence of the objective situation. Specifically, we show that participants vary significantly in their interpretations of staying at home and, thus, also in their experience of boredom. While the non-bored participants interpret the situation as a relief or as irrelevant, the bored participants interpret it as a major restriction that only some are able to cope with.

## 1. Introduction

After the COVID-19 pandemic emerged in December 2019, boredom followed in its wake. While boredom is a well-known personal phenomenon across the lifespan of many people [1] and has also been studied intensively during the last twenty years, if not longer, the pandemic has significantly increased attention on this topic. Suddenly, boredom was everywhere, in traditional as well as social media, (even more) research studies, and everyday conversations. Especially the sudden media coverage implied that the pandemic has caused boredom for numerous people. Indeed, studies have shown that boredom increased significantly during the pandemic containment measures [2,3,4,5,6]. This was no surprise, considering the backdrop of far-reaching everyday life restrictions. Closed recreational opportunities, limited personal contacts, a prescribed home office, and, especially, quarantine and isolation made the everyday lives of many individuals more isolated and monotonous—which are well known crucial drivers of boredom [3,7,8,9]. However, at the same time, it is important to note that boredom is not an inevitable outcome of an objective situation [10,11,12]. Thus, in our study, we focused on subjective interpretation processes instead of objective situations. Specifically, we asked persons under quarantine how they interpreted their situation and analyzed how these interpretations were interwoven with (non-)boredom.

It is already known from previous research that the development of boredom does not depend on the existence of a certain situation alone but also on, for example, the coping strategies applied to a situation [13,14], certain personally traits conceptualized as boredom proneness [15,16], or the ability to remain attentive [11]. In addition, more generally, researchers into emotion who employ social constructivism [17,18] and appraisal theory [19,20,21] have long recognized that the subjective interpretation of a situation is a key factor in the emergence of an emotion. According to these researchers, strictly speaking, it is not a situation but the subjective interpretation of the situation that causes an emotion to follow. Empirical and theoretical research suggests that this also holds true for boredom.

For example, social constructivist Peter Conrad [17] states that boredom is an “interpretive category” and points out that it marks the “endpoint of a subjective interpretation.” He argues that “boredom is not a characteristic of an object, event or person but exists in the relationship between individuals and their interpretation of their experience.” Although this means that boredom is a personal phenomenon to a large extent; it is not determined by certain personality traits only; and it is better understood as a mixture of personal, cultural, and situational influences that inform a person’s perspective (for further explanation, see the next section on social constructivism). In addition to Conrad, other researchers on boredom who employed social constructivism, such as Musharbash [22], Wagner and Finkielsztein [23], and Ohlmeier et al. [24] have empirically shown that the meaning, experience, and causes of boredom are interpretative in nature. Specifically, they emphasize that boredom is culturally or systemically situated. Similarly, but with slight differences, social psychologists also emphasize the importance of subjective interpretation. Under the term “cognitive appraisal theory,” they claim that subjective appraisals by an individual in response to stimuli in the environment is an important and distinct component of emotion [19,20,21]. Psychological boredom researchers who explicitly used appraisal theory suggest that boredom is associated with a perceived lack of meaning [25,26] or a perceived lack of control and value [27].

However, previous research on boredom focused almost exclusively on the situation, the person, and the person–situation fit to explain the emergence of boredom [28]. To date, subjective interpretation processes that assign a certain meaning to a situation have been largely neglected in the research on boredom, with the few exceptions mentioned above. Similarly, while there is already an established body of research that have studied boredom related to the pandemic, none of the research has focused on the underlying interpretation processes that lead to boredom. Instead, most of the studies attribute boredom to situational and demographic factors [2,3,4], focus on the consequences of boredom (e.g., asking whether boredom leads to rule breaking) [29,30,31], or are a combination of all [32]. While investigating the importance of subjective interpretation processes is certainly not new in research on emotion, it is usually not the focus or used productively in boredom research. This means that, in previous research projects, the situation and the interpretation of the situation are rarely differentiated, although they should not be considered interchangeable. Therefore, we hypothesized that subjective interpretation processes may be a so a long-neglected piece of the puzzle of understanding why quarantine or isolation leads to boredom for some but not for others.

Thus, this study aimed to address this research gap. In our qualitative interview study on boredom under quarantine or isolation, we intended to elucidate the importance of perception and interpretation processes in the development of boredom. In our interviews with various people in situations known to easily induce boredom—such as quarantine or isolation—we were able to establish that interpretation processes are a crucial part of the puzzle of boredom. Through a qualitative comparative analysis of our empirical data, we found that quarantine measures were interpreted differently by those who were bored and those who were not. In the following sections, we aim to present quarantine from the perspective of our interviewees. We illustrate step by step how and why they interpret and assess the situation in the way they that they did and how that impacted their sense of boredom. After establishing an interpretative theoretical framework and explaining our methods, we present our results by contrasting the bored participants with the non-bored ones. Specifically, we show how our interviewees interpreted the quarantine or isolation measures in extremely different ways along the following typology:(1)The non-bored: The profiteers, who interpreted quarantine as a relief, andThe indifferent, who interpreted quarantine as irrelevant.(2)The bored:The proactive, who interpreted quarantine as a challenging restriction, andthe resigned, who interpreted quarantine as an overwhelming restriction.

Our results contribute to the literature on boredom per se by focusing on the importance of subjective interpretation processes and also on experiences of quarantine and isolation. Because boredom under quarantine may have significant negative effects for the individual and for collective well-being, this topic is of high importance. Although the issue of boredom might seem trivial against the backdrop of the serious physical health impacts of COVID-19, it really is not. It is increasingly recognized that boredom is not to be taken lightly in terms of public health; it has been defined as an aversive state [11], deeply connected to psychological and physical well-being. It has been associated with tiredness, passivity, stress, restlessness [33], and a slowed-down perception of time [34] and is often accompanied by feelings of entrapment [35], hopelessness, and meaninglessness [25,26]. The long-term consequences of boredom on health and well-being can be serious; it has been correlated with depression, anxiety, loneliness, anger, and aggression [36,37]; linked to an increased risk of accidents at work [38]; and linked to poor social relationships [16]. In addition, there is a connection between boredom and dysfunctional behaviors such as gambling [39], eating disorders [40], and excessive alcohol consumption [41,42]. Regarding the COVID-19 pandemic, there is some evidence that boredom also reduces compliance with social-distancing requirements [29,30,31]; however, others studying the relationship between boredom and compliance have come to the conclusion that there is no relationship between boredom and rule-breaking in real-life situations [32]. All these findings emphasize that it is crucial to understand the drivers of boredom as a prerequisite to developing adequate measures to address it.

## 2. Theoretical Frame: An Interpretative Approach to Boredom under Quarantine

In this study, we investigated boredom through a theoretical lens based on the basic ideas of social constructivism and appraisal theory. Although there are some differences between these approaches, there is an important overlap: both approaches emphasize the significance of subjective interpretations instead of claiming “objective realities” as the originators of emotions.

In general, appraisal theories of emotion propose that situational appraisals are key to understanding an emotion [19,20,21]. For example, depending on the context and person, a monotonous situation may be interpreted as a much-needed break or a lack of a challenge. Whereas the first appraisal is a component of an emotional episode of relaxation, the latter is a component of an emotional episode of boredom. In this view, excluding other components such as action tendencies, physiological responses, behavior, and feelings, an appraisal (e.g., lack of challenge) is a distinct characteristic of an emotion. Beyond regarding appraisal as part of an emotion, some appraisal theorists claim that appraisals can *cause* emotions. They state that appraisals of a situation, instead of the situation per se, cause an emotional or affective response. Thus, in the previous example, it is not the monotony but the interpretation of monotony as a lack of a challenge that leads to boredom. While there is some on doubt whether appraisals are a part of or the cause of an emotional episode [43], appraisal theorists generally agree that situational assessments are important in the understanding of the development and experience of emotions. Traditionally, this insight has been used productively in research into stress [44,45,46,47,48]. For example, in his classic transactional stress model, Lazarus states that stress results from an interplay of different situational appraisals and coping strategies. In his model, stress occurs only when a situation is interpreted as a challenge, danger, or potential loss (first appraisal) and when a person assesses their own resources to be insufficient (second appraisal). If, instead, a person perceives a potentially stressful situation positively or as irrelevant, they will not experience stress [46,47].

Similarly, but with slight differences, social constructivists highlight the importance of subjective interpretations. However, they take the basic idea even further. Social constructivism is a sociological meta-theory that can be traced back to Berger and Luckmann’s groundbreaking work [49], which claims that all reality is socially constructed. Thus, not only the interpretation of a situation (monotony = break vs. monotony = lack of challenge) but also the emotions are constructed. More specifically, social constructivists who are interested in emotions emphasize that “cultural ideologies, beliefs, and norms, as they impinge on social structures, define what emotions are to be experienced and how these culturally defined emotions are to be expressed” [18]. For example, in sociological boredom research, the extent to which boredom is a product of a capitalist society is frequently discussed [50,51] As sociologist Gardiner put it: “boredom has become increasingly recognised as a critical concept that centres on issues and problems of experiencing meaning under the conditions of modern and contemporary society.” In a capitalistic society that worships constant growth, acceleration, and efficiency, boredom is consequentially framed as a state of ”having nothing to do” and considered to be a problem instead of being addressed as an issue of meaninglessness. Thus, how and where we draw boundaries between emotions (“this is boredom” vs. “this is relaxation”) depends on culturally predefined concepts of emotions instead of being an innate physiological process [18,52]. In addition, in social constructivism, there is a strong emphasis on how society influences (constructs) our interpretations and emotions. Consequently, for social constructivists, subjective interpretations are not private but socially grounded and formed by prevailing norms and cultural rules. Conrad [17] notes: “When I lived in Indonesia, people often had to wait for long stretches in the bank or in a government office; Americans would get bored and impatient with the waiting, while Indonesians would see it as something to be expected or as an opportunity to socialize.“ While social constructivism usually focuses on cultures and how they define and conceptualize emotions (or other patterns of understanding), we can also apply this way of thinking to individuals. This is often done in interpretative qualitative research, using, for example, interviews. Through the lens of constructivism, we are able to understand how individuals construct an emotion and then reconstruct the underlying interpretation processes (shaped by society) that lead to an emotion.

In this paper, instead of strictly following either social constructivism or the appraisal theory, we refer to the common idea of both theories: subjective interpretations play a crucial role in the emergence of an emotion. In accordance with social constructivism, we also assumed that those interpretations are grounded in social norms and cultural rules. With this lens, we chose a rather understudied approach to boredom. Hitherto, functionalist [53,54,55], cognitive–affective [11,56], and meaning-based [25,26,57] approaches are far more widespread. While all of those approaches highlight important facets of boredom, they unilaterally reflect a psychological and quantitative research tradition (at least the last two). Thus, their domination reflects the historic imbalance between the (quantitative) psychological approaches and (qualitative) approaches of the social sciences and humanities in boredom research [58,59].

This approach had implications throughout our analysis, specifically on how we understood quarantine or isolation in the context of this study. While often used interchangeably, isolation and quarantine have slightly different meanings. According to the Robert Koch Institute, quarantine is defined as the temporary isolation of someone suspected of being infected or who may be carrying or shedding the virus because of contact with an infected person or cross-border travel in high-risk areas [60]. Persons who are not fully vaccinated and under quarantine usually have to stay at home for 14 days after the last contact with a person with COVID-19.

Isolation, in contrast, is defined as the temporary isolation of someone who has been diagnosed with COVID-19 [61]. People who are in isolation usually have to stay at home for at least 10 days in a specific “sick room” or area and use a separate bathroom (if available). In either case, the local public health authorities order these measures officially. They also prescribe for how long a person has to rem in in isolation or quarantine. While those “official” definitions of quarantine and isolation informed our approach, particularly our sampling strategy (see below), we were open to accepting differential interpretations of quarantine and isolation from our participants.

Specifically, we were interested in the interpretations of situations where participants were required to stay at home for a period of time without being able to go outside or receive visitors—whether this was due to official quarantine, isolation, or even self-imposed restrictions. The terms they used to describe those situations were irrelevant to the study, as we concluded that the official name was not crucial to our interviewees and is even relatively often unknown. Therefore, rather than working with fixed definitions of quarantine and isolation, we investigated what “staying at home” meant to our interviewees. Accordingly, in the following sections, we use the term quarantine* as a catch-all for all situations in which people decided to or were forced to stay at home because of COVID-19.

In line with the prior arguments, we used the idea of “boredom as a result of subjective quarantine* interpretations” as a theoretical heuristic for our empirical data. We specifically searched for overarching interpretation patterns of quarantine* that relate to boredom (or the absence of boredom) in our data. Furthermore, we attempted to explain how those interpretation patterns are shaped by prior experiences and cultural norms.

## 3. Method

To gain in-depth insights into the subjective interpretations of quarantine* and highlight major differences in the perception of it, we conducted an explorative qualitative interview study to answer the questions of (1) “how do persons under quarantine* assess their situation?” and (2) “which underlying interpretation patterns led to those assessments?” For this purpose, we interviewed 13 participants who had been in quarantine or isolation. As a sampling strategy, we used the principle of maximum contrast and analyzed it thoroughly with the type-building qualitative content analysis according to Kuckartz [62]. In the following section, we describe the procedure in detail.

### 3.1. Sampling and Recruiting: “Boredom Contrasts” as Guiding Principle

We interviewed 13 participants who had been under quarantine or in isolation at home for at least 10 days due to (potentially) being infected with the 2019 novel coronavirus. To simplify sampling, we included only individuals who had been officially quarantined or isolated to ensure that all had a joint baseline experience of having had to stay at home. The participants were recruited in two German communities that had been significantly affected by the COVID-19 outbreak shortly after a public festival was held. The first community (community A) is in one of the first German cities that reported a severe outbreak of COVID-19. Thus, most participants from this community experienced quarantine or isolation during February–March 2020. The second community (community B) had its peak COVID-19 outbreak in October–November 2020; thus, most participants had been quarantined at that time. The research design was conceptualized as a retrospective study. The interviews were conducted between June 2020 and March 2021.

The sampling procedure was led by the principle of maximum contrast [63]. We aimed to include participants who had experienced varying degrees of boredom instead of varying the sample with regard to sociodemographics. The reason for this procedure was twofold: (1) with such a small sample size, representativeness is not achievable and (2) we aimed to explain boredom from an interpretative perspective. From this perspective, boredom is not induced by the “objective features” of a situation or person, but by “mental differences,” such as subjective interpretations and resulting behaviors. To establish those differences, it seemed reasonable to vary the outcome (boredom) and not the sociodemographics, which may or may not lead to boredom.

Participants were recruited via two gatekeepers. In community A, our gatekeeper was a former resident who initiated a recruitment call via personal contacts (using the snowball sampling method), Facebook groups, and a supermarket notice. In community B, our gatekeeper was a doctor who asked her COVID-19 patients whether they would consent to participate in the study. We were explicitly searching for bored participants in both cities; however, ultimately, more non-bored people answered our call.

### 3.2. Participants

We eventually recruited four bored and eight non-bored persons, as well as one person (Interviewee E) who was not bored during his first quarantine but was bored during his second quarantine. As Table 1 shows, the sample included different personal backgrounds and quarantine conditions.

### 3.3. Data Collection: Semi-Structured Interviews

We used semi-structured in-depth interview guides according to Helfferich for data collection [64]. In line with the qualitative research approach and the nature of an explorative study, the questions were as open as possible. Thus, we started our interviews with an open narrative stimulus by asking “I would like to go back to the very beginning of your quarantine. Could you please narrate how it all started, how you experienced it, and how it ended?” This stimulus was supposed to encourage our interviewees to establish their own relevance instead of only reacting to the researchers’ assumptions. Once the interviewees completed their narration, we followed up with questions related to their narrative (such as “you mentioned x, could you perhaps explain this further?”) and only then asked specific questions about boredom. If boredom was not addressed by the interviewees, follow-up questions, such as “looking back now on the entire quarantine period, can you recall a situation in which you were bored? (if yes) how did it happen and how did you deal with it? (if no) how do you explain that you were not bored during that time?” were asked. In addition, in a closing phase, we encouraged interviewees to share important situations or thoughts on boredom and quarantine that had not been captured in the interview as yet. With this structure, the interviewees were able to introduce boredom when and how they wanted to, and, at the same time, the interviewer ensured that they talked about boredom—or the absence thereof.

Due to the COVID-19 restrictions, all interviews were conducted via telephone. Eleven interviews were conducted by the first author, Silke Ohlmeier (M.A. Sociology), and the remaining two interviews were conducted by the second author, Isabell Schellartz (M.Sc. Health Economics). Both interviewers are experienced in qualitative interviewing because of their education and practice.

### 3.4. Data Analysis: Qualitative Content Analysis to Build a Typology

The transcribed interviews were analyzed to construct a typology using the qualitative content method described by Kuckartz [62]. The analysis consisted of two major phases.

In the first phase, we developed our categories according to the Kuckartz method of content structuring [62]. After some initial work, such as reading all interviews carefully, highlighting meaningful passages, writing memos, and summarizing each interview, we established a code system. This process was guided by our heuristic idea of boredom as a result of subjective interpretations. Thus, we used the terms “quarantine* assessments” and “quarantine* interpretations” as deductive categories and main frame. To complete our main categories, we inductively searched for subcategories and other major themes beyond our presumptions. After some passes, group discussions, and adaptions, we created a code system (Table 2) and coded all the data.

The data was analyzed by SO as the first coder and CK as the second coder, and they established the code system. Moreover, the material was presented and discussed in two group sessions with researchers from various backgrounds, such as medicine, psychology, sociology, and public health.

In the second phase, we focused on type building. For this purpose, we followed Kuckartz for polythetic type building [62]. Polythetic types are natural types that are inductively extracted from the empirical material. Thus, the types are similar but not completely identical. To construct different quarantine* types (with an emphasis on interpretations), we created a short profile of each interviewee on the basis of our code system. Thereafter, we compared the profiles in regard to similarities and differences, especially related to boredom and subjective quarantine* interpretations. We identified four different interpretations of quarantine*, which are presented in the following section.

### 3.5. Ethics

The study received a positive ethics vote from the Ethics Committee of the University of Cologne (Faculty of Humanities) before it was initiated. All participants were informed about the study aim and protocol and signed a written consent form.

### 3.6. Author Positionality and Reflexivity

The study was mainly devised and conducted by the first author, who is a sociologist and qualitive researcher. It is part of her dissertation on “Social dimensions of boredom,” which mainly focuses on societal influences on boredom. The study was supported by a research team to allow for critical discussion of methodological and interpretative decisions. Up to the time of the completion of the paper, none of the team members had been in quarantine or isolation.

## 4. Results: (Absence of) Boredom and Its Underlying Interpretation Patterns

In our analysis, we identified people who were bored and those who were not. Two main types could be identified in both groups, based on the underlying patterns of the interpretation of quarantine*. We found the non-bored to include the profiteers (Section 4.1.1) and the indifferent (Section 4.1.2), and the bored included the proactive (Section 4.2.1) and the resigned (Section 4.2.2). Each subtype could be characterized by a specific pattern of interpretation of quarantine*, which either promotes or prevents boredom. In the following subsections, we describe and explain each type and the underlying interpretation pattern. Each subsection starts with a comprehensive overview of the main characteristics of the specific type to aid the understanding of the subsequent explanation.

### 4.1. The Non-Bored

We identified two different types in the group of non-bored interviewees, namely, (Section 4.1.1) the profiteers (see Box 1), who interpreted quarantine* as a gain, specifically, as a break from a stressful life, and (Section 4.1.2) the indifferent (see Box 2), who interpreted quarantine* as irrelevant.

Box 1The Profiteers.
**Main characteristics of *the profiteers*:**
Assessed quarantine* positively (enjoyed it);Focused on gains;Felt accelerated or overworked before quarantine*;Interpreted quarantine* as a relief;not bored at all.

**Illustrating anchor quote:**
“If I’m completely honest, I enjoyed this time here at home, as a time to come to rest, because I had a very stressful time in the office before.”**Assigned interviewees:** A, D, F, K

Box 2The Indifferent.
**Main characteristics of *the indifferent*:**
Assessed quarantine* neutrally;Focused on continuations, downplayed restrictions;Valued indoor activities that suited the situation;Interpreted quarantine* as irrelevant;Not bored at all.

**Illustrating anchor quote:**
“Well, I really didn’t mind. It didn’t limit me in anything.”**Assigned interviewees:** B, G, J, M

#### 4.1.1. “Finally, I can…”—The Profiteers: Quarantine* as a Relief

For the profiteers, one characteristic was striking; they were not only not bored, but they even stated that they enjoyed staying at home:

*If I’m completely honest, I enjoyed this time here at home as a time to come to rest, because I had a very stressful time in the office before. So, for me, that was a positive time regarding rest and stress; that was actually good for me, yeah. So, I can’t say that I sat here and didn’t know what to do. I read a book, I started knitting (...) and then at some point, when we were feeling better, we got out board games and played*.(Interviewee K)

These interviewees related quarantine* and isolation to their “busy” (working) lives. Against this background, they interpreted staying at home as a gain of self-determined time that they could fill with purposeful activities, such as spending time with their family, recovering from sickness and stress, and doing housework left undone. Time in quarantine* was perceived as an opportunity to do what they wanted or needed to do anyway but would generally not have the time for. While their pastimes differed, on an abstract level, all described a similar experience; for them, staying at home was a paradoxical experience of finding freedom in spatial limitations.

Particularly the contributions from those who perceived quarantine* as an opportunity to spend time with their families highlighted their working life before quarantine* as a reason for their perspective. Their quotes vividly reveal the strenuous relationship between career and family commitments, which seems to be the basis for their positive evaluation of quarantine*. Juggling work, leisure, and family often left them feeling exhausted and never having enough time for any of their commitments. From this point of view, it is clear why they interpreted the restriction as a relief. It allowed them to concentrate fully on only one of their commitments and to focus exclusively on their families and partners, as the following quotes demonstrate:

*In retrospect, I had an extra month of parental leave. I wouldn’t have had the time otherwise because normally I go to work at half past eight and am usually at home a quarter past six. (…) And no one can take away the time with my daughter from me now. So, for this purpose, in quotation marks, it was good for me*.(Interviewee A)

*You simply took more time for communication with each other than you do when you leave the house in the morning, go to work, and then come back in the evening and are tired and have to cook and have to go shopping. All these things were gone, the shopping was left at the door, you were not stressed, and you took more time for each other*.(Interviewee F)

However, the focus did not necessarily shift to families or persons in relationships. For some, quarantine* simply provided them with more time for recovery and self-care, which was equally valued against the backdrop of the busy life they were temporarily removed from:

*Before, when I drove to the office, I left at seven a.m. and came home sometime in the evening. The housekeeper took care of the dog, then massive stress at work. If you had enough time, you stuffed food inside yourself; if you had time, you drank something. If not, you didn’t. Here, it’s really different. I have a big coffee machine here; I always drink water in between; I value eating in peace and quiet. That didn’t exist before*.(Interviewee D)

Whereas, on the surface, the participants quoted above saw this experience as a relaxing break from a capitalist time regime (such as measurement and commodification of time, temporal expansion, acceleration, appropriation of the future, and unequal temporal autonomy), sublimely, their positive experiences are embedded in capitalist temporalities, especially in the ideas of “using time” and “saving time” as, for example, the following quote from interviewee F illustrates.

*We also did things that we needed to do anyway on the weekend, and by doing this we might have created some space for later. (…) I mean things like painting the floor, painting the bathroom. All the things we [had] postponed for two years already, and then said “come on, we’re doing it now,” and we even have the time for doing it, and still all of us are healthy, and it’s not even vacation that is lost for it. I just say this now, right. Or a weekend that’s wasted. Instead, it’s a time when you normally should have been at work. It is quite a good feeling to be able to use the time for yourself*.(Interviewee F)

The interpretation of quarantine* as “extra time” is based on two aspects of capitalist temporalities:

On the one hand, interviewee F conceptualizes time as something that can be owned, sold, and traded. In accounting for her time, she mentally separates work and leisure. Drawing on a common perception of labor as sold time, work decimates valuable time, which is time that can be used freely for oneself. In this model, time is always a scarce resource because “having time” refers exclusively to non-working time. This is the origin of the need for protecting, saving, and using time. Normally, due to the scarcity of time, non-work time is perceived as precious (it should not be wasted or lost to household work). In this logic, quarantine* is a time gain that is well-used by spending time with the family. However, it can also be used as an investment in the future.

The concept of “extra time” is, therefore, based on the idea that time can be appropriated in a way that it can be planned, predicted, calculated, and forecasted. Accordingly, extra time can be saved for future use, which implies an underlying belief in a linear, predictable future. However, strictly speaking, a future in which interviewee F has to work does not exist, but is only anticipated. Analogously to this, time can only be saved if it is seen as a context-free, standardized measurement unit; if one believes that what one saves now will be worth the same later. This assumption is connected to an old capitalist promise that stated that, if one works enough today, one will be happier tomorrow. The fact that interviewee F is doing housework to create “extra time” later indicates that she has internalized the capitalist idea of being able to invest in a better future.

In summary, quarantine* was interpreted by this group of participants as an opportunity for a recreational break from a busy life, a “time gain” to be either spent now or in the future. Accordingly, they see themselves as profiteers of the quarantine* regime they were subjected to.

#### 4.1.2. “I Still Can…”—The Indifferent: Quarantine* as Irrelevant

Among our interviewees, not all people had strong feelings regarding quarantine* or assessed it as good or bad; two of the interviewees, M and J, assessed it neutrally. Thus, we named this group *the indifferent*. Interviewee M, a retired woman in her 80s and an ideal type of this group, illustrates how some of the indifferent individuals focused on the continuation of their normal everyday routines instead of the limitations:

*The only thing that bothered us was the psychological strain: quarantine, what is still to come? What can we expect? Are we infected? This was the stress. Everything else continued as usual. We stayed at home. We did what we always do*.(Interviewee M)

Participants in this group often construed quarantine* as an invisible and, consequently, irrelevant limitation. They structured their days in a way that quarantine* did not impose any restrictions on their lives and, therefore, became invisible. For this purpose, they relied on long-established, routine recreational activities, such as reading, listening to music, or crafting. Thus, they perceived quarantine* as a less important interruption of their regular lives and a “natural” fit to their preferences and routines. As interviewee J put it:

*It did not limit me as much as others in my environment. Many of them were really limited because they weren’t able to drive out every day. Well, I really didn’t mind. It didn’t limit me in anything, except that I couldn’t do grocery shopping. But I don’t like that anyway*.(Interviewee J)

When asked how she interpreted this difference between her and others, she added:

*This is because, generally, I’m alone quite often, even when I’m healthy. And I can deal very well with myself alone in my house*.(Interviewee J)

Interviewee M is also a person who has cultivated listening to music and reading books as recreational activities for a long time. Accordingly, one could say that her preferences were simply a better fit with the enforced quarantine* regime. However, viewing the quarantine* situation as nothing out extraordinary and, consequently, stoically continuing her recreational activities helped her to keep negative feelings (including boredom) at bay. When asked about restlessness, interviewee M said:

*It was felt in rudiments that I was too restless to listen to music. However, if I listened to music anyway, I noticed that I became calmer and that it fell off me a bit. So, I can really recommend to anyone to just try it. Sometimes you don’t feel like it, but I thought, music always helped me; it will help me now, too, and then it was like that*.(Interviewee M)

Therefore, apart from a “natural fit” of personal preferences and situational freedoms, there was an active effort on the part of the participants of this group to maintain a particular mindset of irrelevance towards quarantine* that guided the structuring of daily activities.

### 4.2. The Bored

We identified two different types in the group of the bored interviewees: (Section 4.2.1) the proactive (see Box 3), who interpreted quarantine* as a challenging restriction, and (Section 4.2.2) the resigned (see Box 4), who interpreted quarantine* as an overwhelming restriction.

Box 3The Proactive.
**Main characteristics of *the proactive*:**
Assessed quarantine* measures negatively;Focused on losses *and* opportunities;Perceived that they were responsible for shaping the situation;Interpreted quarantine* as a challenging restriction that could be endured with something;Experienced periods of severe boredom.

**Illustrating anchor quote:**

*„So, you have to do– I think I have to do something to actively beat boredom, be it with thoughts or with actions, but I HAVE to do something.“*
**Assigned interviewees:** C, H

Box 4The Resigned.
**Main characteristics of *the resigned*:**
Assessed quarantine* measures neutrally to negatively;Focused exclusively on losses;Perceived themselves as a victim of external circumstances;Interpreted quarantine* as an overwhelming restriction that they had no means to withstand;Felt severely and persistently bored (at least when external circumstances were difficult).

**Illustrating anchor quote:**

*„…you’re fine, but you’re not allowed to go out, you’re not allowed to do anything. And then you just think, yes, what you COULD do, but you don’t really feel like doing anything.”*
**Assigned interviewees:** E, I, L

#### 4.2.1. “Eventually, I’ll Find Something New”—The Proactive: Quarantine* as a Challenging Restriction

We identified interviewees H and M as one subtype of the bored, the proactive. Compared with the profiteers and the indifferent, these participants believed that being in quarantine* was extremely difficult. They characterized it as a painful loss of purposeful activities, such as going to work, spending time with family and friends, or participating in social activities. Thus, interviewee M (who has chronic obstructive pulmonary disease and was severely ill with COVID-19 so decided to self-quarantine after being in official isolation) described her situation as a big hole:

*Nothing is the way it was. All the things; I went to demonstrations. I’m 63 now, almost 64. I’ve always been an active person. I went to dance school in spite of my wheel walker. I’ve always taken my music lessons. Once a week, I take classical singing lessons. I’ve always participated in public discussions. I did all that before. I CAN’T anymore. That’s ALL fallen away. There’s a big, big hole in my life*.(Interviewee C)

Moreover, interviewee H, who was isolated from the rest of his family, found his situation extremely challenging. When asked whether he would change something if it happened again, he stated:

*I said once, I’ll never go alone again in the basement in the sense of I would somehow try to separate it differently spatially, because of this very lonely time there. ALL alone was really difficult*.(Interviewee H)

Consequently, these participants experienced boredom during quarantine*, especially at the outset. When asked about boredom in contrast to her regular life, interviewee M stated:

*Much more boring. The quarantine time is much more boring, yes. (...) But it has already gotten better. It’s not quite as bad anymore*.(Interviewee C)

Similarly, interviewee H responded as follows when asked whether he could remember boring situations:

*Actually, yes. Especially in the beginning, it was just extremely difficult*.(Interviewee H)

However, despite all the difficulties, the proactive interpreted their isolation as a restriction that they could endure; a situation that was manageable despite all the difficulties. In contrast to the resigned (Section 4.2.2), they perceived having to stay at home as a great challenge, but not more than a challenge. After all, a challenge can be overcome. This interpretation goes hand in hand with a specific perspective of boredom and also of their own agency, namely that boredom is a consequence of one’s own actions rather than of external restrictions. When asked about boredom, the proactive referred to their own mindset and behavior.

For example, interviewee C reflected on her own responsibility in being bored, even explicitly shifting from a general “one has to” to a personal “I have to”:

*So, you have to do—I think I have to do something to actively beat boredom, be it with thoughts or with actions, but I HAVE to do something. If you don’t do that—if I don’t do that, I sink into nirvana*.(Interviewee C)

Moreover, she was convinced she could work on her boredom:

*It [being bored] doesn’t have to stay that way if you work at it; I’m convinced of [it]. If I keep working on it and try to open doors, I can fill that hole again. But it takes time. I don’t think it’s an unchangeable situation; I don’t think that*.(Interviewee C)

Thus, with believing in their own abilities and focusing on opportunities instead of losses, they were able to adapt to the situation. Eventually, both interviewees found ways to compensate for their losses and cope with their boredom:

*The television was on in the background all day. Yes, I think that was kind of the reason why I started checking social media more often when there was a hot topic on TV. Then you could ask on the social networks and see what other people thought about it. That’s how it started, yes... later, I just tried… I interviewed myself and asked myself, “what would you now recommend to someone else who is not doing so well?” (...) And then I was able to find answers here or there and to pick out the right thing. And then I forced myself to implement it*.(Interviewee C)

*Then we had the idea of playing games on Skype and so on, for example, Monopoly—I was in the basement rolling the dice. And the family executed the moves upstairs, so to speak, that I ordered from downstairs. That was… a lot of ideas were then there, later. Then the boredom was okay. I also then had a dartboard in the basement so that I could play darts in my free time and so on. So this was okay then*.(Interviewee H)

In summary, the proactive interpreted quarantine* as a challenge but not a breakdown. It created difficulties and losses, but these can be overcome. This interpretation helped them to activate resources—internal and external—to manage the situation in a way that makes it bearable for them.

#### 4.2.2. “I Can’t Anymore…”—The Resigned: Quarantine* as an Overwhelming Restriction

As a second subtype of the bored, interviewees I, L, and E were identified as *the resigned*. They experienced the stay-at-home measure as an overwhelming restriction, that is, a restriction that they had no means to cope with and that could not be changed for the better, which eventually led to resignation. In contrast to the proactive, who were able to manage the situation to prevent or reduce boredom, for the resigned, the feeling of being bored did not change throughout the duration of quarantine* but only lifted once it was over. While both interviewees shared the interpretation pattern “overwhelming restriction,” their focus differed. Whereas interviewee I perceived quarantine* as a non-shapeable situation, a total restriction, interviewee L felt that quarantine* was being forced on her and, therefore, imprisoned her, which made it unbearable and left her in a constant state of powerlessness.

##### Quarantine* as a Non-Shapeable Situation

One of these three interviewees, interviewee I, especially experienced quarantine* as an overwhelming restriction that inevitably led to boredom because she interpreted the situation as non-shapeable. In a way, she framed it as a total restriction that does not allow any freedom for purposeful activities:

*Well, the first day, you have to deal with yourself; you’re really thinking about what happens now and so on and then I just noticed it’s not getting worse; I’m still fine and then actually you’re fine, but you’re not allowed to go out, you’re not allowed to do anything. And then you just think, yes, what you COULD do, but you don’t really feel like doing anything*.(Interviewee I)

For her all the options for activities to do indoors were “not the same” in quality and quantity. For example, while the proactive interviewees eventually started to communicate with others online or via telephone and experienced some relief from boredom, she used telephone or messenger services for pure information exchange and insisted that communication via those tools could simply not compare to the way it was before (and would be after):

*As I said, the [communication via WhatsApp] was just a “I’d like to come down now, will you go upstairs or to the garage?” or “dinner’s ready.” (...) As I said, you talked to people on the phone, but that’s not the same as doing something with someone in person*.(Interviewee I)

Similarly, playing with her dogs while being prohibited from going for a walk with them did not satisfy her at all:

*We then went behind the house with them, but that is also not something you can entertain yourself with. Then you have thrown a ball two-three times, but otherwise you could do nothing there and as I said, if you are normally used to go out with them every day, in the morning an hour and a half and at noon half an hour, then there is simply something missing*.(Interviewee I)

During her quarantine*, she could not even concentrate on the activities that she usually liked that were still available to her.

*Reading and such, I normally do like very much, but simply could not concentrate duringn that time. That actually intensified my boredom very much because every activity, if you have done something for a while and you are completely alone, then it is no fun*.(Interviewee I)

Eventually, with all opportunities for purposeful activity perceived as restricted, all she could do was wait for sleep to come:

*You get used to it […] I was just tired a lot and then I just lay down in bed, and at some point, I fell asleep and then you slept for an hour and a half and then another hour and a half passed*.(Interviewee I)

Thus, when asked whether there was anything good about quarantine*, she answered:

*No, I actually didn’t like anything at all. Except the last day, when you knew—when you got the call that tomorrow the quarantine is over*.(Interviewee I)

A similar yet slightly different attitude was expressed by interviewee E who also perceived quarantine* as a non-shapable situation that has the potential to induce boredom but does not necessarily have to. While not actively working on shaping the situation, he had the good fortune to be in quarantine* with his girlfriend and her mother, who supported him through it. While rather passive in his argumentation, he highlighted the availability of support (parents close by) and the circumstances being right (nice weather) but concluded that quarantine* is a situation that one must accept and endure:

*It was April; weather was fine; the property behind the house is big. It was like a vacation at home. No, but it was okay. Weather was fine, we spent time outside. Overall, it was not too bad. Parents did the grocery shopping, we cooked nice food and drank coffee, whatever. So, it was really bearable, and you can’t change it anyway*.(Interviewee E)

However, while he was fortunate with the circumstances during his first quarantine* and did not experience boredom, this changed when he was quarantined* for a second time. His girlfriend was not with him; the weather was bad; and, with most chores having been done already, he had no means to endure the quarantine*:

*It was quite a bit more annoying, simply for the reason that my girlfriend did NOT have to go into quarantine with me, and, of course, I was happy for her. However, it is different when you see that your partner can just do whatever she wants, and you sit around feeling stupid. Also, the weather was not that nice anymore, and then I really had cabin fever*.(Interviewee E)

##### Quarantine* as a Forced Restriction

The other interviewee (interviewee L) from the group of the resigned did not suffer from perceiving the situation as non-shapeable per se but more generally because being ordered to stay at home was an externally determined decision that she did not agree with. For her, quarantine* was a restriction imposed by others and forced upon her, a situation comparable to that of a prisoner, which she found difficult to accept. Being resistant to the situation disturbed her concentration notably and prevented her from actively shaping it. As illustrated in the following text from her interview, the impact of subjective interpretation is evident. Interviewee L is the ideal type of a person who does not suffer because of the spatial restriction per se but because of the meaning they ascribe to it. In her interview, the emphasis on the coercive nature of quarantine* is particularly communicated in the criminal metaphors that she used, for example, when describing her contact with the public health department:

*They had already called me in the morning and told me that I should please stay at home and then also added the threat: “If we find out that you are not at home, there will be a fine of there will be a fine of 1000 euros.” The whole procedure was a bit—yes, I felt like a criminal*.(Interviewee L)

The fact that she was convinced that she was not infected with SARS-CoV-2 aggravated the situation. However, even after two negative PCR tests, she was not allowed to leave quarantine*—because of one positive rapid test that had led to the whole situation in the first place.

*A few days later, I was informed that I had been tested negative at my family doctor. Nevertheless, I had to stay in quarantine because the rapid test was positive. I must say that I didn’t understand that for a long time. I then voluntarily had another PCR test done by the doctor, and that was also negative, but it didn’t change anything*.(Interviewee L)

Therefore, tor her, the situation was comparable to that of a wrongly convicted prisoner, which left her feeling extremely externally determined and stigmatized. As she said, the problem was not actually having to stay at home but being forced to do so (for no good reason). Comparing the experience of self-quarantine with quarantine* because of the official order, she concluded:

*And then I just realized–in retrospect, I think the quarantine period was so bad for me because it was ordered. I’ve noticed that now. Those first eleven days that I quarantined myself like that, they weren’t so bad. But this pressure, “You mustn’t;” it triggered me a lot, even up to crying fits*.(Interviewee L)

Reading the situation as being externally enforced left her with a feeling of powerlessness, and she saw no opportunities for positive change. Instead, it rendered her restless and she felt caught up in her own thoughts:

*[…] and then now comes the time when I was alone. […] You’re lonely, but you can’t use the time—so alone. You can’t use the time because your head is going crazy and there are too many thoughts in your head*.(Interviewee L)

Notably, she reflected on the time being “crazy” because so much of it was just “in her head.” However, those thoughts and interpretation patterns were powerful in that they rendered activities normally enjoyed undoable.

*What struck me in retrospect was that I couldn’t do what I normally like to do because I might not have had so much time, because there was so much pressure. It was as if I had said to myself, “No, you’re allowed to do that now, too.” As I said, it was quite crazy in my head*.(Interviewee L)

To summarize, this case was particularly interesting because it powerfully demonstrates how the reading of a situation as an (unjustly) imposed restriction more than a spatial limitation as such had a significant impact on how (boring) quarantine* was perceived to be.

## 5. Discussion

Our results suggest that the bored and non-bored interviewees differed significantly in their interpretations of quarantine*; whereas the non-bored interpreted quarantine* as a relief or as irrelevant, in the interviews, the bored ones perceived it as a major restriction. Furthermore, the data indicate that, depending on how the bored individuals positioned themselves (as shapers of the situation vs. victims of circumstances), they were either able or unable to cope with boredom. Regarding the situation through the lens of these differing self-images, restrictions were either perceived as a manageable challenge, or they were overwhelming and led to persistent boredom. Thus, according to our data, boredom is related to subjective interpretations in two ways; it is a consequence of situational interpretations as well as interpretations of the self. In line with the basic ideas of constructivism [17,18] and appraisal theory [19,20,21,46], our findings could lay the groundwork for the insight that subjective interpretations, and not only the situation, are crucial to the emergence of boredom.

This result from the present study differs from most other common approaches to boredom, such as the situation- or person-based approach, and varies slightly from person-by-situation-based approaches. The next section provides a short overview of the different approaches to explain the emergence of boredom and explains why the additional perspective on interpretative patterns adds important insights.

In contrast to the traditional-situation-based approach that claims that boredom is the result of monotony, repetition, or under-stimulation [8,65,66], we believe that this view alone is inadequate [10]. Instead, we argue that, depending on a person’s current needs, monotony or under-stimulation can be as much a relief as it can be boring. Despite the fact that our interviewees had been in similar situations, our study suggests that the monotony of having to stay indoors did not lead to boredom for everyone; it was not even perceived as monotony by all interviewees. Thus, with our research, we add more differentiation to the existing studies that have shown correlations between pandemic restrictions and boredom [2,3,4,32]. While it is, of course, important to recognize that the pandemic, quarantine, and isolation increased boredom for many people, it is also important to explore and understand the influence of subjective interpretations in the background and how that may make a difference in the perception of the situation.

This study is also different from person-based approaches to boredom and might help to add nuance to those analyses. Person-based approaches argue that personal characteristics are a key cause of boredom [11,28]. For example, psychoanalysts such as Fenichel [67] or Greenson [68] emphasize that boredom is a result of an individual’s inability to articulate his or her desires. Other person-based researchers also highlight cognitive influences, such as difficulties in concentrating [10,69,70]. On the basis of our results, we argue that it does not do justice to the prevailing complexity to focus on only the person. To explain boredom, one has to consider other influencing factors. In our opinion, an interpretative approach reflects this complexity insofar that interpretations implicitly include personal, situational, and cultural influences. As our research suggests, interpretations are influenced by the person, situation, and additional circumstances. Indeed, they are related to the person but not as fixed as personal characteristics or personality traits. Personality characteristics and traits most likely impact but not predispose the interpretation of a situation. For example, the profiteers’ interpretation of quarantine* as a relief is also rooted in a certain work culture and significantly colored by the experience of being overworked before. Another influencing factor is the capitalist time regime [71] that is part of all the participants’ socialization. The subjective interpretations of the individuals might have differed if they were in different life stages or in another time or culture.

Finally, our approach also differs slightly from approaches that focus on the person-by-situation fit. Such studies emphasize the interaction between a situation and person and presume that boredom arises from a mismatch between personal characteristics and situational demands [10,72]. We found some support for this claim in our study. Specifically, individuals who perceived quarantine* as irrelevant profited from a good person-by-situation fit. Because they enjoyed pursuing indoor activities, being in quarantine* was easier for them. However, with our approach, we focused on the relationship between interpretations and person-by-situation fit instead, and we want to add the following remarks to the commonly used approach. Although a good person-by-situation fit can promote certain interpretations (for example, interpreting containment measures as irrelevant is more likely if one’s interests and hobbies involve indoor activities), as our study indicates, this is also valid if the situation is reversed. For some of our interviewees, believing that restrictions are irrelevant became a self-fulfilling prophecy, which led them to behave in accordance with their beliefs and interpretations. In this sense, interpreting quarantine* as irrelevant is not simply the product of a natural fit but also the result of an active effort to maintain a particular mindset. Similarly, on the other side of the spectrum, those who feel that all purposeful activities take place outdoors (as is true for the resigned) shifted the focus to what was lacking and blocked the process of working actively on the situation (indoors).

Furthermore, our approach supports various other empirical findings on boredom; for example, that boredom is associated with feelings of entrapment [35] or a perceived lack of meaning [25,26]. In addition, there are important intersections with studies that focus on classroom boredom. Similar to quarantine or isolation, students who experience boredom in the classroom have to cope with externally determined spatial limitations. Therefore, we may be able to learn what will support students to cope with boredom of those studies. In accordance with what our research suggests, studies on class-related boredom among students have shown that cognitive reappraisal strategies (looking for positive aspects of the situation) help to overcome boredom [73]. In addition, studies on class-related boredom suggest that boredom can be reduced by providing individuals with more autonomous support and increasing the perceived value of what they are doing [74].

In our opinion, going beyond the established approaches and broadening the view on boredom through the suggested perspective can reveal hitherto overlooked paths for interventions. Whereas boredom-inducing personality traits (e.g., introversion/extroversion or sensation-seeking) are difficult to change, subjective interpretations are aspects of boredom emergence that a person can actively work on. In inescapable situations (such as quarantine*), reflecting or working on perceptions may be the only solution to prevent or overcome boredom. Thus, in our opinion, explaining boredom through subjective interpretations is a missing piece of the puzzle, and it complements the already existing approaches to understanding boredom. However, our intention is not to reinforce a discourse that shifts the responsibility for emotions to people (and, therefore, possibly away from policymakers who manage circumstances) or induce pressure on individuals to experience quarantine* as something positive or productive. As helpful as it might be to work on interpretations, it is equally important to accept negative feelings, such as boredom, as part and parcel of quarantine or isolation. Experiencing emotion, as unpleasant as it might be, can also be the start of or part of a productive coping process, and may act as an incentive to become politically active and work for necessary change on the meso- or macro-level. Thus, while embracing the opportunity to identify avenues for combating boredom, the primary aim of our study is to create awareness of the complexity of quarantine* boredom and, thereby, advocate for thinking that moves beyond the presumption that overcoming boredom is a simple problem that can be solved by finding something to do.

## 6. Limitations and Future Research

Due to the qualitative research design, this study did not investigate a sociodemographic representative sample but exploratively focused on the relationship between boredom and subjective interpretations of quarantine*. Thus, following the interpretative research paradigm of qualitative research, we did not make any claims regarding the frequency or distribution of boredom. Instead, it was our goal to examine *how* a phenomenon (boredom) interacts with another one (interpretation patterns of quarantine*) in our particular cases [75]. Thus, while we did not investigate the influences of sociodemographics, we were able to reconstruct important interpretation patterns that interact with boredom on an individual and group level. Conducting a relatively small qualitative study provided us with the opportunity to gain in-depth insights and remain open to the relevancies of the interviewees. While this is a key strength of this study, every research design has certain innate drawbacks.

As this is a qualitative study using type-building content analysis, we worked with a relatively small sample size and did not include all possible quarantine* situations systematically. While we included different degrees of boredom and worked with maximum contrast, we did not aim for explicit theoretical saturation. However, for example, as Guest et al. [76] showed in an experiment, saturation can occur within the first six to 12 interviews. There is no consensus among qualitative researchers on the question of “how many interviews are enough.” An adequate sample size depends on the research aims and objectives, traditions within epistemic communities, and the available time and resources [77]. Given that resources and time are usually limited, a larger sample size often means going broader instead of deeper (which produces different but not necessarily better results). Still, a larger qualitative or quantitative study could be useful to ensure that the capture of all possible boredom dimensions and interpretation patterns and to validate the results. The present explorative study could be used as a solid foundation for such a purpose.

Moreover, this study could possibly have benefited from a gradual sampling strategy to deepen our understanding of inductively emerged categories (such as the relevance of the time before quarantine* and self-perceptions). However, because we decided to complete our interviews before analyzing the data, we did not have an opportunity to adapt our interview guide with regard to such themes.

Furthermore, due to our sample size and analysis method, we could only demonstrate which appraisals (positive, neutral, boring) were associated with which interpretation patterns, and, thus, we cannot claim any causal relationship. To better understand how appraisals and interpretation patterns are related, a qualitative grounded theory study may be helpful.

Finally, with the focus on the subjective interpretations of having to stay at home, other influences on boredom were excluded. For example, in our study, we did not explicitly examine the influence of time when the interviewees were in quarantine* (and related fatigue effects), specific household constellations, weather conditions, or living environment. It is important to note that this a consequence of our research focus and does not mean that subjective interpretations constitute the only factor that influences boredom. As previous research on boredom has shown, boredom is influenced by the situation, personal characteristics, and/or the person-by-situation fit. Indirectly, some of these factors were captured through the lens of subjective interpretations since many of the interviewees referred to them when explaining their perspectives. Nonetheless, to examine the underlying situational and personal factors systematically and to understand how these intervene with subjective interpretation, more research is required.

Please note that our goal did not include explaining the causes of quarantine* boredom in an all-encompassing manner. Instead, our study aimed to show what an interpretative approach would reveal and should be understood as a complementary instead of a competing approach.

## 7. Conclusions

While it is important to acknowledge that quarantine or isolation induces boredom for many people, one should be wary of the details. Our findings suggest that differentiating between situations and interpretations of situations is crucial for understanding boredom under quarantine or isolation. Within our interviews, subjective interpretations differed from perceiving quarantine or isolation as a relief or as irrelevant for the non-bored individuals to perceiving it as a major restriction for the bored individuals. Thus, our research indicates that, for a comprehensive understanding of boredom, it is also important to consider the impact of interpretations.

## Figures and Tables

**Table 1 ijerph-19-02207-t001:** Sample Overview.

.	Gender	Age	Occupation	PCR Test Result	Duration	Symptoms *	Living Situation **	Boredom
A	M	31	Bank clerk	Positive	4 weeks	Mild	Family	No
B	M	26	Department manager	Negative	5.5 weeks	Mild	Alone	No
C	F	63	Not working due to COPD	Officially negative, self-belief: positive	14 days officially, then self-quarantine	Severe	Daughter	Yes
D	M	55	Property manager	Negative	10 days	None	Son	No
E	M	21	Trainee; sales	Not tested	2 × 14 days	None	1. Partner + mother-in-law 2. Alone	1. No, 2. Yes
F	F	46	Kindergarten teacher	Negative	10 days	None	Family	No
G	F	37	Teacher	Negative	14 days	None	Separated	No
H	M	39	Instructor; factory	Positive	4 weeks	Mild	Separated	Yes
I	F	53	Waitress	Positive	10 days	Mild	Separated	Yes
J	F	49	Ambulant care	Not tested	17 days	Severe	Alone	No
K	F	58	Clerk in hospital	Positive	4 weeks	Severe	Partner	No
L	F	59	Pediatric nurse (retired, marginal employment)	Officially positive, Self-belief: negative	10 days self-quarantine, 14 days isolation	None	Alone	Yes
M	F	83	Retired	Negative	8 days	None	Partner	No

* Mild symptoms indicate symptoms of a cold; severe symptoms include fever, pain, exhaustion, breathing difficulties, being bedridden. ** Alone indicates that the person was alone in the house or flat; separated indicates that the person stayed in a separate room or cellar away from partner and family who were in the same house; partner includes husband, wife, girlfriend, or boyfriend; family includes partner and children under the age of 18.

**Table 2 ijerph-19-02207-t002:** Overview of the Code System.

**Quarantine* assessments:** ○ Positive; ○ Neutral; ○ Boring: ▪ Temporary/partial; ▪ Persistent. **Quarantine* interpretations:** ○ Gain/relief; ○ Irrelevant; ○ Major restriction: ▪ Challenging restriction; ▪ Non-shapable situation. **Time before quarantine*** **Self-perception**

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
