# Peer review of "Having a Break or Being Imprisoned: Influence of Subjective Interpretations of Quarantine and Isolation on Boredom"

_ijerph, 2022, doi:10.3390/ijerph19042207_

Round 1

Reviewer 1 Report

I first want to share that I am a quantitative researcher by nature and qualitative methodology is not my area expertise. I share this so it can be considered in light of my responses and understanding of the results. The study was unique and timely in considering the interviews of individuals (in social isolation/quarantine) during the pandemic and how their responses, particularly about boredom, can be interpreted with two theoretical lenses (cognitive appraisal theory and social constructivist theory). I think there are some interesting findings to be presented, however, I recommend some major revisions pertaining to the paper organization and interpretation/presentation of findings. Please see my recommendations below.

Structure

It would help to add some more subheadings for the organizing of the methods. For example, Participants, Procedure (which includes recruitment/data collection), and perhaps a researcher positionality statement. A clear methodology statement and objectives statement (research question) would be helpful. It would benefit the paper to have clear sections so you can find these important parts.

Please articulate the main goal of the study earlier in the introduction.

Are all the interview questions included in the methods?

Interpretation/Presentation of results

I think the authors may be overgeneralizing/overstating their findings beyond what the results allow them to say. Given the interviews are based on 13 individuals, the study should be more about the individual level, not group level. I would revise the discussion to be closer to the results of the study (example – use more exploratory language, less causal inference, for e.g., “our findings could lay the groundwork for…” and “these findings suggest” could help.

A suggestion: the authors could draw on control-value theory in their interpretation of the findings. CVT deals with several dimensions (valence, activation) as well as control and value appraisals that fit nicely in your findings and could strengthen your study's empirical implications. It also brings in social context/situation plus appraisals as impacting/driving emotions.

A potential limitation for the study concerns the participants experiencing social isolation in March 2021 may have burnout from the pandemic (compared to those who had to do it earlier). This could possibly have an impact on the experience of boredom. Did you find more "bored participants in the later interviews?

The authors discuss themes like capitalist time regime in interpretation of results but this is not introduced in the intro/background/theoretical argumentation. I think clearer connections could be made in this regard.

Minor

In the definition of quarantine, it also can be issued for international/cross-border travel (p4).

A careful read through of the manuscript again for small typos would be helpful (e.g., page 1: “for a several people”, citation in middle of word subjective; and page 2: “or a lack of perceived lack of control and value”).

Was there any other demographic info collected - ethnicity, education, that could be added to the Method?

On page 9, the illustrating anchor quote and quote in text are redundant.

I don't think you need to use an asterisk for quarantine each time after explaining what it refers to. This is just my preference/recommendation.

On page 12, the main characteristic of the proactive – “partly severely” seems conflicting.

Footnote 3 - what does this mean "his overall attitude"?

On page 17 - I didn't see any reference to person-based approaches. Are the authors meaning person-centered (aka latent class and latent profile analytic type methods)?

Page 19 -I like the intention-to bring preliminary awareness for the complexity of quarantine.

Author Response

Dear Reviewer,

Thank you for giving us the opportunity to revise our paper. We appreciate your helpful comments. We have now revised the manuscript following these comments. All changes are made using the track change mode for your convenience. Below you find a point-by-point response to all comments raised. We hope that the manuscript is now acceptable for publication.

Please note: Unfortunately, it was not possible to add any references in track change mode. When we tried to, track change mode switched off and our changes were not visible anymore. Hence, for now we decided to put new references in brackets and will change it to the proper citation style after all changes have been seen and accepted by you and other reviewers.

Comments

Answers

Reviewer 1

I first want to share that I am a quantitative researcher by nature and qualitative methodology is not my area expertise. I share this so it can be considered in light of my responses and understanding of the results. The study was unique and timely in considering the interviews of individuals (in social isolation/quarantine) during the pandemic and how their responses, particularly about boredom, can be interpreted with two theoretical lenses (cognitive appraisal theory and social constructivist theory). I think there are some interesting findings to be presented, however, I recommend some major revisions pertaining to the paper organization and interpretation/presentation of findings. Please see my recommendations below.

Thanks for your appreciation of our paper!

It would help to add some more subheadings for the organizing of the methods. For example, Participants, Procedure (which includes recruitment/data collection), and perhaps a researcher positionality statement. A clear methodology statement and objectives statement (research question) would be helpful. It would benefit the paper to have clear sections so you can find these important parts.

We rearranged the methods section and added information to allow for greater clarity in the presentation of methods. However, we didn’t change the subheading “sampling and recruiting” to procedure since it’s a common heading in qualitative research.

In addition, we added a positionality statement under the subheading “research team and reflexivity”.

Please articulate the main goal of the study earlier in the introduction.

For better readability, we now spell out the research question in the first paragraph.

Are all the interview questions included in the methods?

We decided to include the remaining two interview questions, which hadn’t been mentioned before (3.3. data-collection). Now all interview questions are included.

I think the authors may be overgeneralizing/overstating their findings beyond what the results allow them to say. Given the interviews are based on 13 individuals, the study should be more about the individual level, not group level. I would revise the discussion to be closer to the results of the study (example – use more exploratory language, less causal inference, for e.g., “our findings could lay the groundwork for…” and “these findings suggest” could help.

We revised the discussion and conclusion in accordance with the suggestions and used more exploratory language.

A suggestion: the authors could draw on control-value theory in their interpretation of the findings. CVT deals with several dimensions (valence, activation) as well as control and value appraisals that fit nicely in your findings and could strengthen your study's empirical implications. It also brings in social context/situation plus appraisals as impacting/driving emotions.

While we believe this is an interesting suggestion, we have decided not to include a discussion of CVT as it addresses so many additional concepts that would have to be explained that it will likely blow the discussion out of proportion. We have prioritized the situating of the findings within the boredom literature as we find this most interesting to the readers of this special issue.

A potential limitation for the study concerns the participants experiencing social isolation in March 2021 may have burnout from the pandemic (compared to those who had to do it earlier). This could possibly have an impact on the experience of boredom. Did you find more "bored participants in the later interviews?

We now explicitly acknowledge the non-consideration of timing in the limitations. Following the qualitative paradigm, we refrain from drawing conclusions on the basis of frequencies. Even if we found more bored participants in later interviews with our sample size we could not generalize this finding. 

The authors discuss themes like capitalist time regime in interpretation of results but this is not introduced in the intro/background/theoretical argumentation. I think clearer connections could be made in this regard.

We didn’t mention this in the introduction because it’s an inductive empirical finding. However, we agree that the point is difficult to understand without theoretical framing. Thus, we introduced the idea of capitalism as a boredom influencing factor in our theoretical frame now. Furthermore, we explained it in more detail in the empirical findings.

In the definition of quarantine, it also can be issued for international/cross-border travel (p4).

We added cross border travelling as a reason for quarantine.

A careful read through of the manuscript again for small typos would be helpful (e.g., page 1: “for a several people”, citation in middle of word subjective; and page 2: “or a lack of perceived lack of control and value”).

We’ve corrected the typos suggested. There will be a final professional proof reading before publication (not done yet due to the time limit for the revision and possible changes in content).

Was there any other demographic info collected - ethnicity, education, that could be added to the Method?

We did not collect any demographic info beyond the info already mentioned in the method section.

On page 9, the illustrating anchor quote and quote in text are redundant.

The illustrating anchor quote is one quote from the text which represents the group best.  It serves the purpose of having a quick overview. Thus, it’s redundant by nature. We understand that it’s irritating because the quote follows directly after the anchor example, but changing this would mean using a less fitting quote. Accordingly, we did not change anything.

I don't think you need to use an asterisk for quarantine each time after explaining what it refers to. This is just my preference/recommendation.

We believe the asteriks helps highlighting and keeps reminding that quarantine in our approach is a social construct. We accordingly did not change anything.

On page 12, the main characteristic of the proactive – “partly severely” seems conflicting.

With the phrases “partly severely” we wanted to emphasize that the intensity of boredom was high, but participants managed to overcome it and/or also had days without boredom (in contrast to the last group). However, we agree that the term “partly severely” is confusing. We changed it to “experienced periods of severe boredom”.

Footnote 3 - what does this mean "his overall attitude"?

We changed the wording to make it more understandable. The sentence now reads  “Nonetheless, we think with his rather passive coping style he fits best in this group”.

On page 17 - I didn't see any reference to person-based approaches. Are the authors meaning person-centered (aka latent class and latent profile analytic type methods)?

 We are not well-versed in the concepts mentioned by the author, but are referring to a different classification that is used among certain boredom researchers. We refer here to the idea that characteristics of the person play a key role in the experience of boredom. In their paper “causes of boredom” Mercer-Lynn et al. (2014) frame these approaches as “person based”. Other approaches center on the situation or the person-situation-fit to explain the emergence of boredom. In this paragraph we want to make clear, that subjective interpretations are not only a question of personal characteristics, but also include situational and cultural aspects and focusing on these interpretations adds to the understanding of boredom. Also, we have added additional references to person-based approaches for easy accessibility at the beginning.

Page 19 -I like the intention-to bring preliminary awareness for the complexity of quarantine.

Thank you!

Reviewer 2 Report

  1. This study focuses on the influence of isolation and quarantine on individual boredom, an important issue during the current COVID-19 pandemic. In turn, individual's cognition and evaluation on their isolation and quarantine affect their boredom level. This research has certain practical significance.
  2. The study is reasonably designed, appropriately conducted, and has reliable results and conclusions, which can provide help and basis for reducing the boredom of quarantined people and improving their health during epidemic prevention and control.
  3. There are still some improvements to be made in this study. For example, the references are not comprehensive enough. Because isolated boredom is similar to classroom boredom, one can't leave and have to stay somewhere. Therefore, more recent literature and empirical studies on classroom boredom can be cited in terms of the antecedent of boredom and boredom intervention. For example, according to studies on class-related boredom, isolated boredom may also be reduced by providing individuals with more caring, autonomous support in a particular environment and increasing their perceived value of what they are doing.

Author Response

Dear Reviewer,

Thank you for giving us the opportunity to revise our paper. We appreciate your helpful comments. We have now revised the manuscript following these comments. All changes are made using the track change mode for your convenience. Below you find a point-by-point response to all comments raised. We hope that the manuscript is now acceptable for publication.

Please note: Unfortunately, it was not possible to add any references in track change mode. When we tried to, track change mode switched off and our changes were not visible anymore. Hence, for now we decided to put new references in brackets and will change it to the proper citation style after all changes have been seen and accepted by you and other reviewers.

Reviewer 2

This study focuses on the influence of isolation and quarantine on individual boredom, an important issue during the current COVID-19 pandemic. In turn, individual's cognition and evaluation on their isolation and quarantine affect their boredom level. This research has certain practical significance.

The study is reasonably designed, appropriately conducted, and has reliable results and conclusions, which can provide help and basis for reducing the boredom of quarantined people and improving their health during epidemic prevention and control.

Thank you!

There are still some improvements to be made in this study. For example, the references are not comprehensive enough. Because isolated boredom is similar to classroom boredom, one can't leave and have to stay somewhere. Therefore, more recent literature and empirical studies on classroom boredom can be cited in terms of the antecedent of boredom and boredom intervention. For example, according to studies on class-related boredom, isolated boredom may also be reduced by providing individuals with more caring, autonomous support in a particular environment and increasing their perceived value of what they are doing.

We agree that findings from classroom boredom can be insightful also for situations of quarantine/isolation. As suggested, we added a paragraph on similarities to classroom boredom to the discussion.

Reviewer 3 Report

The authors examined via qualitative interviews the situational interpretations that might lead some but not others to perceive isolation (as a function of the pandemic) to be boring. I find the work interesting and timely, but I do have some concerns that I hope the authors can address.

The claim made early in the introduction that boredom "became" ubiquitous as a function of the pandemic is overstating things. Chin et al (2017) Bored in the USA highlights the ubiquity of the experience across the lifespan in a large experience sampling study.  And while boredom was perhaps understudies in the 1990's there has been building research from many labs over the past 20 years - so I think this claim ought to be toned down somewhat and Chin et al ought to be cited.  

The authors frame their work in terms of a social constructivist account of boredom - but by my reading of the literature the more dominant accounts of boredom are functionalist (Bench & Lench, 2013; Elpidorou, 2014, 2018, 2020) or cognitive-affective (Eastwood, et al., 2012; Tam et al., 2021) or meaning based (Van Tilburg & Igou, 2011; Westgate & Wilson, 2018 and others refs). Much of that research is not cited and I think some argument as to why the social constructivist view is preferred here is warranted.   

In highlighting that Westgate and colleagues find no relation between boredom and rule breaking (ref 29) the authors are blurring the distinction between state boredom (measured by Westgate et al) and trait boredom proneness (measured by Wolff et al and Boylan et al who did find an association between trait boredom and rule breaking in the pandemic). Two things could be done her - omit the Westgate reference or preferably, highlight the distinction between trait and state more clearly.   

I am unfamiliar with the qualitative approach taken here - but from the outset a very small n of 13 is concerning. If I take social constructivist views on face value - how many distinct social influences might impinge on a sample of 13? SES might differ vastly, family/household make-up likely differs, caregiving duties will be important. I am concerned that with such complexity not much can be said on the basis of 13 people. Hopefully the authors can argue their case against this concern. I see the authors make some steps towards this already - but I'm not overly convinced. Sure, one could select based on outcome - but that doesn't address the concern that 13 is still not a large n. The authors state that this can't be representative (I agree) but then what do we learn?  If there are 13 distinct interpretations we learn nothing, but even if there are only two, we're still left wondering whether these two are the dominant interpretations.  This all becomes even more problematic when we discover that the sample is skewed towards the non-bored. 

Content analysis could be more rigorously done using unsupervised machine learning algorithms such as structural topic models (e.g., Hu, N., Zhang, T., Gao, B., & Bose, I. (2019). What do hotel customers complain about? Text analysis using structural topic model. Tourism Management72, 417-426.) although here again the small n may pose problems. 

Why weren't coders who were blind to the experimental aims used (they could be trained on the coding system developed)?

I see that the majority of the results is spent in detailed qualitative analysis of responses, but I am stuck on the sample size - with only 2 and 3 in the separate "bored" groups I just don't know what we can learn. Happy to have the authors try to highlight to me that I am wrong - especially since qualitative interview work of this kind is not within my area of expertise. 

Minor points:

Some editing might be warranted - e.g., "Closed recreational offers.." - should that be "opportunities"?  And later there is an errant"{Citation}" in the middle of a word. 

Also, the phrase "a lack of perceived lack of control..." needs editing - I think the authors simply mean a "perceived lack of control".

Author Response

Dear Reviewer,

Thank you for giving us the opportunity to revise our paper. We appreciate your helpful comments. We have now revised the manuscript following these comments. All changes are made using the track change mode for your convenience. Below you find a point-by-point response to all comments raised. We hope that the manuscript is now acceptable for publication.

Please note: Unfortunately, it was not possible to add any references in track change mode. When we tried to, track change mode switched off and our changes were not visible anymore. Hence, for now we decided to put new references in brackets and will change it to the proper citation style after all changes have been seen and accepted by you and other reviewers.

Reviewer 3

The authors examined via qualitative interviews the situational interpretations that might lead some but not others to perceive isolation (as a function of the pandemic) to be boring. I find the work interesting and timely, but I do have some concerns that I hope the authors can address.

We believe we can.

The claim made early in the introduction that boredom "became" ubiquitous as a function of the pandemic is overstating things. Chin et al (2017) Bored in the USA highlights the ubiquity of the experience across the lifespan in a large experience sampling study.  And while boredom was perhaps understudies in the 1990's there has been building research from many labs over the past 20 years - so I think this claim ought to be toned down somewhat and Chin et al ought to be cited. 

Although we do believe that the attention for boredom was severely increased by the pandemic, we agree that the introduction was probably overstating things. As suggested, we toned it down and cited Chin et al. (2017)

The authors frame their work in terms of a social constructivist account of boredom - but by my reading of the literature the more dominant accounts of boredom are functionalist (Bench & Lench, 2013; Elpidorou, 2014, 2018, 2020) or cognitive-affective (Eastwood, et al., 2012; Tam et al., 2021) or meaning based (Van Tilburg & Igou, 2011; Westgate & Wilson, 2018 and others refs). Much of that research is not cited and I think some argument as to why the social constructivist view is preferred here is warranted. 

The reason we chose social constructivism as our theoretical perspective is precisely because it is not so well established in the field and viewing the phenomenon through this lens will add to the understanding of boredom. We believe that the reason it has so far not been used widely is not because it is not helpful, but for reasons of the disciplinary traditions of psychology (where boredom is mostly studied). We added our argument on page 4 to allow for more transparency.

In highlighting that Westgate and colleagues find no relation between boredom and rule breaking (ref 29) the authors are blurring the distinction between state boredom (measured by Westgate et al) and trait boredom proneness (measured by Wolff et al and Boylan et al who did find an association between trait boredom and rule breaking in the pandemic). Two things could be done her - omit the Westgate reference or preferably, highlight the distinction between trait and state more clearly. 

We highlighted the distinction more clearly now. 

I am unfamiliar with the qualitative approach taken here - but from the outset a very small n of 13 is concerning. If I take social constructivist views on face value - how many distinct social influences might impinge on a sample of 13? SES might differ vastly, family/household make-up likely differs, caregiving duties will be important. I am concerned that with such complexity not much can be said on the basis of 13 people. Hopefully the authors can argue their case against this concern. I see the authors make some steps towards this already - but I'm not overly convinced. Sure, one could select based on outcome - but that doesn't address the concern that 13 is still not a large n. The authors state that this can't be representative (I agree) but then what do we learn?  If there are 13 distinct interpretations we learn nothing, but even if there are only two, we're still left wondering whether these two are the dominant interpretations. This all becomes even more problematic when we discover that the sample is skewed towards the non-bored.

The reviewer refers to the long-standing debate between quantitative and qualitative researchers. Quantitative and qualitative research start from different premises and pursue very different goals. In difference to quantitative research, qualitative research does not want to produce representative results or even make any claims regarding distribution or dominance of views. Qualitative researchers rather provide overviews of possible perspectives, in-depths analysis of experiences or theories of processes (that can and should be tested in the following by rigorous quantitative analysis, but that was not our goal here). In our study, we were trying to provide an overview of the breadth of interpretations of quarantine/isolation and how those pertain to boredom. Clearly, there are other factors than quarantine interpretations that influence whether boredom develops, but as we were not aiming to make any quantitative claims, we were not controlling for those.

With regard to the sample size: Qualitative studies generally work with small samples as they aim for depth instead of breadth and sample sizes ranging from 10-20 interviews are the norm. There is again a long-standing debate on the question “how many interviews are enough” and studies have shown that saturation can be reached with a sample of 6-12.

We now elaborated even further on the goals and limitations of our (qualitative)research in the limitations sections, so that it’s more transparent to researchers who are not familiar with qualitative research designs. Specially, we added more information and references on the question “how many interviews are enough”. We hope that we could thereby address the concerns of the reviewer.

Content analysis could be more rigorously done using unsupervised machine learning algorithms such as structural topic models (e.g., Hu, N., Zhang, T., Gao, B., & Bose, I. (2019). What do hotel customers complain about? Text analysis using structural topic model. Tourism Management72, 417-426.) although here again the small n may pose problems. 

Why weren't coders who were blind to the experimental aims used (they could be trained on the coding system developed)?

We believe that the comment is based on a misunderstanding of the goals and premises of (certain schools of) qualitative research. In our understanding, blind coding and machine learning can serve an important purpose in ensuring objectivity in quantitative research which can, however, not be uncritically adopted for qualitative research. Quality criteria in qualitative research differ from those of quantitative research in that subjectivity is often seen as a resource for than a challenge to doing high-quality research. Here, it’s also important to understand that there are many different research traditions within the field of qualitative research. While some approaches (as cited by the reviewer) are closer to quantitative thinking, others (like our approach) are more interpretative in nature. In our approach, two persons coming to a different coding system doesn’t mean that one must be wrong. There can be two different interpretations of the data, which both are right (just reflecting different aspects, different questions or different theoretical framings). Thus, it is considered more helpful to make transparent how one comes to an interpretation and reflecting one’s own preassumptions than doing bling coding and using machine learning. Accordingly, we believe that our analysis could not have profited from an employment of the tools suggested by the reviewer, but would indeed have been detrimental as machine learning is not yet able for the kind of in-depth analysis we were aiming for.

I see that the majority of the results is spent in detailed qualitative analysis of responses, but I am stuck on the sample size - with only 2 and 3 in the separate "bored" groups I just don't know what we can learn. Happy to have the authors try to highlight to me that I am wrong - especially since qualitative interview work of this kind is not within my area of expertise.

We really appreciate that the reviewer is candid about his/her quantitative background. We agree that qualitative research operates from a very different background which makes the communication across the “disciplinary divide” difficult sometimes. We hope that we were able to explain better where we come from – also for possible readers via our additions to the methods and limitation sections.

Some editing might be warranted - e.g., "Closed recreational offers.." - should that be "opportunities"?  And later there is an errant"{Citation}" in the middle of a word. 

Also, the phrase "a lack of perceived lack of control..." needs editing - I think the authors simply mean a "perceived lack of control".

We corrected the mentioned issues. A detailed English proofreading will be done before publication (not done yet due to the time limit for the revision and possible changes in content).

Round 2

Reviewer 1 Report

I think the authors have done a good job addressing my concerns. Well done. I did notice some inconsistent reporting of in-text citations throughout, but as the authors note they had issues with the tracked changes function and would fix the issue at the next stage. 

Reviewer 3 Report

The authors have done a concerted job of attending to my concerns - and I think they're right that this is a difference of methodologies at play. But I really appreciate the extensive efforts to clarify and make transparent the issues. So, I am happy for the paper to be published in its current form.